# Exploring Neurophysiological Mechanisms and Treatment Efficacies in Laryngeal Dystonia: A Transcranial Magnetic Stimulation Approach

**DOI:** 10.3390/brainsci13111591

**Published:** 2023-11-15

**Authors:** Maja Rogić Vidaković, Joško Šoda, Joshua Elan Kuluva, Braco Bošković, Krešimir Dolić, Ivana Gunjača

**Affiliations:** 1Laboratory for Human and Experimental Neurophysiology, Department of Neuroscience, School of Medicine, University of Split, 21000 Split, Croatia; 2Signal Processing, Analysis and Advanced Diagnostics Research and Education Laboratory (SPAADREL), Faculty of Maritime Studies, University of Split, 21000 Split, Croatia; jsoda@pfst.hr; 3Piedmont Neuroscience Center, Oakland, CA 94610, USA; joshuakuluvamd@piedmontneurosciencecenter.com; 4Department of Otorhinolaryngology, University Hospital of Split, 21000 Split, Croatia; bboskovic01@gmail.com; 5Department of Interventional and Diagnostic Radiology, University Hospital of Split, 21000 Split, Croatia; kdolic79@gmail.com; 6Department of Radiology, School of Medicine, University of Split, 21000 Split, Croatia; 7Department of Biology and Human Genetics, School of Medicine, University of Split, 21000 Split, Croatia

**Keywords:** laryngeal dystonia, transcranial magnetic stimulation, motor evoked potential, cortical silent period, corticobulbar tract, corticobulbar motor evoked potentials, thyroarytenoid muscle, cricothyroid muscle, whole-genome sequencing, laryngeal vibration, deep brain stimulation, ENT

## Abstract

Laryngeal dystonia (LD), known or termed as spasmodic dysphonia, is a rare movement disorder with an unknown cause affecting the intrinsic laryngeal muscles. Neurophysiological studies point to perturbed inhibitory processes, while conventional genetic studies reveal fragments of genetic architecture in LD. The study’s aims are to (1) describe transcranial magnetic stimulation (TMS) methodology for studying the functional integrity of the corticospinal tract by stimulating the primary motor cortex (M1) for laryngeal muscle representation and recording motor evoked potentials (MEPs) from laryngeal muscles; (2) evaluate the results of TMS studies investigating the cortical silent period (cSP) in LD; and (3) present the standard treatments of LD, as well as the results of new theoretical views and treatment approaches like repetitive TMS and laryngeal vibration over the laryngeal muscles as the recent research attempts in treatment of LD. Neurophysiological findings point to a shortened duration of cSP in adductor LD and altered cSP duration in abductor LD individuals. Future TMS studies could further investigate the role of cSP in relation to standard laryngological measures and treatment options. A better understanding of the neurophysiological mechanisms might give new perspectives for the treatment of LD.

## 1. Introduction

Laryngeal dystonia (LD), known or termed as spasmodic dysphonia, is a type of focal dystonia affecting the intrinsic laryngeal muscles during speech [1]. LD is a rare, idiopathic disease with unknown exact incidence or prevalence. It has been reported that the prevalence of LD is up to 35 per 100,000 people in the general population [1]; however, in some countries, the prevalence was reported to be lower (i.e., Iceland and Japan). Similar to other forms of isolated or focal dystonia (e.g., hand dystonia, cervical dystonia, blepharospasms, and oromandibular dystonia), the pathophysiology of LD is regarded as multifactorial. Recent reports define dystonia as a neural network disorder with alterations in brain structure, function, and neurotransmission [2,3,4]. The basal ganglia with thalamus and cerebellum are regarded as the core of structural and functional disorganization, while reduced availability of striatal dopamine D_2_/D_3_ receptors decreases the inhibitory activity within the indirect basal ganglia pathway, leading to hyperexcitability of the thalamus [2,4].

The two most common types of LD are adductor LD and the relatively rare abductor LD. The condition is characterized by task specificity and selective speech impairment with preserved whispering, breathing, laughing, crying, and yawning [1]. Adductor LD is characterized by a strained-strangled voice quality with intermittent voice breaks during vowel production (phonation task), while abductor LD is characterized by intermittent breathy voice breaks occurring predominantly on voiceless consonants [1]. However, some patients may exhibit a combination of both adductor and abductor spasms, leading to a mix of strained and breathy voice qualities. Caucasians and women are significantly more affected, while the average age of symptom onset is 40 years [1]. Symptoms can appear gradually, but a sudden onset is recorded in 45% of cases [5]. In 82.4% of cases, the disorder remains focal, but patients may later develop other dystonia types, such as cranial or generalized dystonia [5]. Approximately 25.3% of LD patients have a positive family history of dystonia, and 11.8% of them have some other movement disorder, indicating a genetic predisposition for LD [1]. Specific extrinsic factors show a correlation with the development of LD. Chronic stimuli of the upper respiratory system in the form of frequent infections, gastroesophageal reflux disease, and inhaled irritants increase the risk of developing LD. Psychiatric conditions such as depression, anxiety, and stress are a risk factor [6]. Approximately 43–46% of LD patients reported emotional trauma before symptom onset. Emotional trauma immediately before symptoms have been found in 43–46% of LD subjects [6]. People who frequently use their voice, e.g., singers and professors, constitute a risk population [1]. The LD diagnosis is made by a multidisciplinary team examination that includes a laryngologist, a speech and language therapist, and a neurologist. Due to the lack of strict diagnostic guidelines and still unknown pathophysiological mechanisms involved in LD, the diagnosis is delayed an average of 5.5 years from the onset of symptoms, and the multidisciplinary team experts agree with diagnosis only in 34% of cases [1]. In the differential diagnosis, one must include other disorders that cause laryngeal spasms, such as essential vocal tremors and muscle tension dysphonia [7,8]. Both voice tremor and muscle tension dysphonia are present in approximately one-third of LD individuals [1]. The main differentiating factor of LD from other diseases is the symptom occurrence during voice production/phonation and speech [1,5]. The standard treatments for LD have typically included botulinum toxin injections, speech/voice therapy, and, in some cases, surgical interventions (ENT and neurosurgical procedures). Botulinum toxin injection into specific laryngeal muscles is the standard primary therapeutic option leading to functional voice improvement and improved voice-related quality of life of LD subjects [9,10]. Botulinum toxin blocks the release of acetylcholine from nerve terminals. Three compounds of botulinum toxin type A (Botox^®^, Dysport^®^, and Xeomin^®^) and type B (NeuroBloc^®^) are commercially available for LD treatment [9]. The recommendation for botulinum toxin injection in adductor LD is the thyroarytenoid (TA) muscle (bilateral or unilateral) with Botox^®^/1–5 MU, Xeomin^®^/10–40 MU, and Dysport^®^ per side, while for treatment of abductor LD the posterior cricoarytenoid (CT) muscle (bilateral or unilateral) is recommended by applying Botox^®^/2.5–5 MU, Xeomin^®^/20–40 MU, or Dysport^®^ per side [9]. Thereby, botulinum toxin causes flaccid paralysis of the muscle about 48 h after injection; however, its effect is transient [9]. A significant proportion of patients require repeat procedures, and some patients might experience transient dysphagia after administration of botulinum toxin injection.

Besides standard treatment management of LD, some individuals with LD explore alternative therapies such as acupuncture, yoga, or relaxation techniques to manage stress, which can exacerbate symptoms. While these approaches may provide some relief, their efficacy is not well-established through scientific research. New theoretical views and treatment approaches like transcranial magnetic stimulation (TMS) [11] and laryngeal vibration over the laryngeal muscles are present in recent attempts in the treatment of LD [12].

This narrative review first summarizes the previous knowledge on TMS methodology for mapping the corticobulbar tract for laryngeal muscles. Secondly, the review aims to present the current state-of-the-art results on the cortical excitability findings in LD acquired with TMS, focusing on the cortical inhibition measure, namely, cortical silent period (cSP). The review discusses the findings of surgical treatments and new research attempts by applying repetitive TMS (rTMS) and laryngeal vibration in the treatment of LD. Additionally, a brief presentation of the outcome measures used in evaluating treatment success is given. Also, a summary of the genetic findings in LD is presented, with future consideration discussed. Finally, future recommendations on using potential subclinical markers to assess LD diagnostics and monitor therapy outcomes will be presented, considering also a new future approach.

## 2. Existing TMS Methodology for Recording the Corticobulbar Motor Evoked Potentials from Cricothyroid and Thyroarytenoid Muscles

The methodology for mapping the primary motor cortex (M1) for cricothyroid muscles (CT) was developed by Espadaler et al. [13], Deletis et al. [14], and Rogić Vidaković et al. [15] using three-dimensional (3D) magnetic resonance imaging (MRI) e-field-navigated TMS (Nexstim Plc., Helsinki, Finland) in healthy subjects. A single pulse is applied to the M1 brain region for recording the corticobulbar motor evoked potentials (CoMEPs). It is important to emphasize that the CoMEP response represents a neurophysiological marker of corticobulbar tract integrity for laryngeal muscles. Therefore, previous findings on TMS mapping of the M1 for laryngeal muscle representation while recording CoMEPs from laryngeal muscles were primarily achieved in the context of preoperative mappings in neurosurgical settings to obtain norms for CoMEP latency in healthy subjects [13,14,15]. To facilitate eliciting CoMEPs before electromyographic activity related to speech onset, the visual object naming task was applied to the subjects while magnetic stimulation was applied at zero time [14]. Later on, the methodology for recording CoMEPs from laryngeal muscles was confirmed intraoperatively by applying direct cortical stimulation (DCS) or transcranial electrical stimulation (TES) over the M1 for laryngeal muscle representation in anesthetized patients during neurosurgical procedures [16,17]. The responses from the CT muscle were obtained from two hook-wire electrodes (the size of 76 μm of diameter passing through a 27-gauge needle) (#003-400160-6, SGM, Split, Croatia) inserted in the muscle. Stimulation over the M1 for CT elicited CoMEPs in the contralateral CT with a mean latency between 11.75 ± 2.07 ms and 12.66 ± 1.09 ms [13,14,15] in healthy subjects. Furthermore, Chen et al. [18,19] recorded the CoMEPs responses from the TA muscle with a 30 mm, 27-gauge needle loaded with a pair of fine-wire hooked electrodes (#019-772800, Nicolet Co., Middleton, WI, USA). A frameless stereotactic neuronavigational system (BrainSight, Rogue Research Inc., Montréal, QC, Canada) mapped the M1 for thyroarytenoid muscles [18,19]. It has been observed that the average CoMEP latencies for left hemisphere cortical stimulation were 15.6 ±2.3 ms in the left TA and 13.1 ± 2.0 ms in the right TA. The average CoMEP latencies for the right hemisphere cortical stimulation were 15.5 ± 2.8 ms and 13.1 ± 2.3 ms for the left TA and right TA, respectively. Figure 1 (left side) depicts M1 for upper extremity representation (abductor pollicis brevis, APB muscle) as a referent positive spot in TMS studies and M1 for laryngeal muscle representation. The single pulse magnetic stimulation over the M1 for laryngeal muscle representation evokes CoMEP response from target laryngeal muscles (Figure 1, on the right side).

### Procedure for Stimulation and Recording of CoMEPs from Laryngeal Muscles

Prior to the mapping of M1 for laryngeal muscle representation, the mapping of M1 for upper extremity representation is performed by determining the resting motor threshold (RMT), MEP latency, and MEP amplitude of the MEP response recorded from upper extremity muscles (i.e., APB). A single magnetic pulse maps M1 for APB and M1 for laryngeal muscle representation. Surface electrodes are used for the recording of MEPs from APB muscles. The distance between the cortical representation for M1-APB and M1 for CT muscle representation is approximately 25.19 ± 6.52 mm, with CT muscle representation lateral to the APB muscle [13]. Slight facilitation is needed to obtain CoMEPs from laryngeal muscles [13,14,15]. As an example, Figure 1 shows CoMEPs elicited from TA and CT muscles by applying supramaximal stimulation intensity of 170% regarding the RMT intensity required to elicit MEPs from the upper extremity APB muscle. Anatomical guidelines for the percutaneous placement of the hook-wire electrodes in the CT muscles are determined by Hirano and Ohala [20]. In order to facilitate electrode placement, it is helpful to palpate the contracted CT muscle belly between the lateral side of the thyroid and cricoid cartilage during the production of a high-pitch sound. The palpation of the contracted CTHY muscle belly can be used as a landmark over the skin and marked with a pen. The electrode is inserted in the CT muscle under an angle of 30–40° [20]. In obese subjects, the CT space as the anatomical reference might be difficult to find [13]. Furthermore, for insertion of the hook-wire electrode into the TA muscle, the skin is pierced in the midline in a sagittal direction. The needle is passed through the cricothyroid membrane directly under the lower border of the thyroid cartilage at an angle of 30° laterally and 15° superiorly. Alternatively, to reduce the risk of entering the airway, the cricothyroid membrane can be penetrated a few millimeters laterally to the side to be examined using the lower lateral entry angle (20°).

## 3. Evaluation of Corticobulbar Motor Evoked Potentials and Cortical Silent Period in Thyroarytenoid and Cricothyroid Muscles in LD

The cortical silent period (cSP) is a non-invasive measure of intracortical inhibition in the M1 associated with laryngeal muscle representation. The cSP is detected as a period of inhibition in the electromyography (EMG) signal following CoMEP response during voluntary facilitation, such as voluntary phonation of the sound/a/. The cSP reflects an intracortical inhibitory process mediated by GABA_A_ and GABA_B_ receptors [19]. Chen et al. [18] reported cSP duration for the TA muscle ranging from 41.7 to 66.4 ms in healthy subjects. The average cSP duration from left hemisphere cortical stimulation was reported to be 53.7 ± 7.8 ms in the left TA and 52.8 ± 7.3 ms in the right TA. Furthermore, the average cSP duration from right hemisphere cortical stimulation was reported to be 53.4 ± 7.8 ms in the left TA and 54.5 ± 5.9 ms in the right TA [18]. Chen et al. [18,19] used single TMS pulses delivered using a 70 mm figure-of-eight coil connected to the Bistim2 and 2002 stimulator set (the Magstim Company Ltd., Whitland, UK). Recently, Konstantinović et al. [21] reported findings of cSP duration in the CT muscle in healthy subjects, distributed from 40 ms to 60.83 ms, and from the ipsilateral CT muscle, from 40 ms to 65.58 ms. The authors used paired hook-wire electrodes with insertion needles of 50 mm in length with a caliber of 7 mm (22 G), a lead wire length of 0.40 mm, and 2 PTFE-insulated stainless-steel wires with a blue touch-proof connector (#003-400162-6, SGM, Split, Croatia), and stimulation was applied to the M1 for CT representation with e-field-navigated TMS (Nexstim Plc., Helsinki, Finland) [21]. Figure 2 depicts reproducible CoMEPs and cSPs recorded from TA and CT muscles. Magnetic stimulation intensity required to elicit CoMEPs and cSP from the laryngeal muscle is supramaximal to the RMT intensity for eliciting MEPs from the APB muscle [13,14,15].

Regarding LD disease, cSP has been investigated in the TA muscle by a single research group and has been reported to be shortened in adductor LD [19]. The duration of cSP was 41.4 ± 9.2 ms from left TA (ipsilateral) and 39.8 ± 9.6 ms from right TA (contralateral) when the left M1 for laryngeal muscle representation was stimulated [19]. When stimulating the right hemisphere for laryngeal muscle representation, the duration of cSP was 39.7 ± 7.7 ms from the left TA (contralateral) and 40.7 ± 5.8 ms from the right TA (ipsilateral) muscle [19]. No changes were observed in the latency of the CoMEPs in LD subjects compared to healthy controls [19]. Recently, a research group reported altered cSP duration in the TA muscle in one individual with the rare abductor LD type [22]. The same research group reported altered cSP duration in the CT muscle for single case reports of adductor and abductor LD subjects [23]. Finally, Chen et al.’s [19] findings on cSP duration in adductor LD point to similar findings of reduced inhibition found in focal hand dystonia, cervical dystonia, and non-affected muscles in LD [4,24,25,26,27,28].

## 4. New Approaches in the Treatment of LD: Repetitive Transcranial Magnetic Stimulation (rTMS) and Laryngeal Vibration

The effects of neuromodulation using non-invasive repetitive transcranial magnetic stimulation (rTMS) are clinically unexplored in LD disease. Since recent findings point to reduced intracortical inhibition in adductor LD [19,26], Prudente et al. [11] applied single-session low-frequency rTMS in seven adductor LD patients and six healthy control subjects. It is known that the low-frequency rTMS is often applied as a single, low-frequency stimulation train over the M1 area, lasting 10 to 20 min. Prudente et al. [11] reported positive phonatory changes (small effect size in decreased phonatory breaks) in individuals with adductor LD. Presently, there is an ongoing interventional study (ClinicalTrials.gov Identifier: NCT05095740) investigating the effectiveness of rTMS in decreasing the overactivation in the TA muscle, designed as a proof-of-concept, randomized study comparing LD subjects receiving rTMS with those receiving sham rTMS. The estimated study completion (ClinicalTrials.gov Identifier: NCT05095740) is reported to be 31 May 2026.

Furthermore, the proof-of-concept study investigated the effect of laryngeal vibrotactile vibration (vibration frequency of 100 Hz) using encapsulated vibro-motors (Model 307-100, Pico Vibe, Precision Microdrives Ltd., London, UK; diameter: 8.8. mm, length 25 mm) positioned percutaneously over the laryngeal area (lateral to thyroid cartilage) on speech quality and cortical activity in LD subjects. A total of 9 subjects out of 13 subjects exhibited a reduction in voice breaks and improved speech quality, with symptom improvement persisting 20 min after applying laryngeal vibration [12]. The study also reported that laryngeal application induced a significant suppression of theta band power over the left somatosensory-motor cortex and a significant increase in gamma activity over the right somatosensory-motor cortex [12].

## 5. Invasive Treatments of LD: Where Do We Stand?

### 5.1. Deep Brain Stimulation (DBS) in Dystonia: Neurosurgical Procedure

Contrary to the non-invasive nature of rTMS, invasive deep brain stimulation (DBS) of globus pallidus internus (GPi) or subthalamic nucleus (STN) has been approved by the FDA for the treatment of drug-refractory generalized, segmental, cervical dystonia, hemi dystonia, and essential tremor [1]. It has to be pointed out that no current DBS findings are reported for isolated focal LD treatment, except for patients with essential tremor and coincident LD in whom DBS of the thalamus’s ventral intermediate (VIM) nucleus was targeted [29,30,31].

In a 79-year-old woman with the essential tremor of the dominant right upper limb and LD, unilateral left thalamic VIM stimulation (DBS on) significantly improved LD vocal dysfunction compared with no stimulation (DBS off), as measured by the USDRS (*p* < 0.01) and voice-related quality of life (VRQOL) (*p* < 0.01) [31]. The patient’s voice was evaluated with the stimulation turned off for 14 days and on for 14 days. The unilateral left electrode’s stimulation parameters were Case +, contacts, 0 off, 1-, 2 off, 3 off, pulse width 90 ms, frequency 185 Hz, and voltage 3.0 V.

Krüger et al. [30] presented an 85-year-old woman, right-handed, with essential tremor of the limbs and LD, and a 73-year-old man with mixed left-handedness with essential limb tremor and LD. The female patient underwent bilateral VIM nucleus DBS, with DBS parameters: left VIM case +, contacts 0 off, 1(−), 2, and 3 off, frequency 185 Hz, pulse width 90 µs, and amplitude 2.8 V; right VIM case off, 8 and 9 off, 10(+) and 11(−)frequency 185 Hz, pulse width 90 µs, and amplitude 2.5 V. The male patient underwent bilateral VIM nucleus DBS with the following parameters: left VIM case (+), contacts 0 (−), 1, 2, and 3 off, frequency 185 Hz, pulse width 60 µs, voltage 1.6 V; right Vim case +, contacts 8 off, 9 -, 10 and 11 off, frequency 185 Hz, pulse width 60 µs, and voltage 2.5 V. In the female patient, the OMNI—Vocal Effort Scale (OMNI-VES) [32] showed the most improvement (80%) with both DBS sides on. With just the left DBS side turned on, a 60% improvement was achieved, whereas only a 20% improvement occurred with just the right DBS side on. The results were similar to the Unified Spasmodic Dysphonia Rating Scale (USDRS) [33] total score and four of the USDRS subcategories (overall, voice arrest, voice tremor, and voice intelligibility). In the mixed left-handed patient, the OMNI-VES also showed that the most improvement (71%) was achieved with bilateral DBS sides on. With the right DBS side on, a 57% improvement was achieved, whereas a 14% improvement occurred with the left DBS side on.

Evidente et al. [29] presented three patients with essential tremor and LD with bilateral VIM nucleus DBS. The first patient was a 74-year-old, right-handed woman with hand tremor and LD. The programming settings were as follows: for the left VIM, settings were case (+), 0(−), 3.9 volts, 90 µs, and 180 Hz; for the right VIM, settings 11(+), 8(−), 3.7 volts, 90 µs, and 180 Hz. The patient subjectively assessed her voice, noting that she could easily phonate with no vocal tremor. The second patient was a 71-year-old, right-handed woman with hand tremor, blepharospasm, and LD, and the programming settings were as follows: for the left VIM, settings were 3(+), 0(−), 2.7 volts, 60 µs, and 180 Hz; for the right VIM, settings were 11(+), 8(−), 2.4 volts, 60 µs, and 180 Hz. The patient subjectively assessed the voice quality as better in the stimulated state. The third patient was a 65-year-old, left-handed man presenting with bilateral hand tremor and LD with the following stimulator settings: for the left VIM, settings were 3(+), 0(−), 3.7 volts, 90 µs, and 185 Hz; for the right VIM, settings were 11 (+), 8(−), 3.7 volts, 90 V, and 185 Hz. The patient reported improved LD symptoms and vocal tremor in the stimulation state.

Lastly, according to the literature findings on 5 September 2023, one observational prospective cohort study is underway (ClinicalTrials.gov Identifier: NCT05506085) to apply DBS of GPi in treating 12 LD individuals. The estimated study completion (ClinicalTrials.gov Identifier: NCT05506085) is reported to be 1 May 2024.

### 5.2. ENT Surgical Procedures in LD Treatment

ENT surgical procedures include selective laryngeal adductor denervation–reinnervation surgery [34,35,36,37], bilateral vocal fold medialization [38], TA muscle myectomy/myoneurectomy [39,40,41,42,43], and type II thyroplasty (TPII) [44,45,46].

Selective denervation disunites the adductor recurrent laryngeal nerve at its insertion into the TA and the lateral CT muscles, with the proximal stump exteriorized to prevent regeneration, and the ansa cervicalis nerve is anastomized to prevent the regeneration of recurrent laryngeal nerve axons to the TA and the lateral CT muscles. Selective laryngeal adductor denervation–reinnervation surgery for adductor LD can provide relief from voice breaks in patients with LD refractory to botulinum toxin injection [34,35]. Inspecting the laryngeal adductor denervation–reinnervation surgery, a total of 43 LD patients’ findings were reported [34,35,36,37].

Regarding the treatment of abductor LD, botulinum toxin injection is often unsuccessful, and surgical options include bilateral vocal fold medialization [38]. Type I thyroplasty is performed under local anesthesia using a silastic implant designed to medialize the vocal folds in six patients with abductor LD [38]. The authors reported a reduction in voice breaks [38].

A bilateral TA muscle myectomy surgical approach under microlaryngoscopy was described by Nakamura et al. [43] in seven patients with adductor LD. The postoperative breathy voice was reported as the disadvantage of the method. Partial myectomy of the TA muscle and neurectomy of the thyroarytenoid branch of the inferior laryngeal nerve resulted in long-term improvement of vocal quality of 15 patients with adductor LD [41,42]. Furthermore, Benito et al. [39] reported the utility of the bilateral posterior CT partial myoneurectomy in the treatment of a single male patient with abductor LD who had no improvement after six botulinum toxin injections over 21 months. During the postoperative period, the patient reported improved quality of life with breathy voice breaks.

A long-term evaluation of thyroplasty type II with titanium bridges revealed patient satisfaction with the postoperative voice status and a significant improvement in VHI sore postoperatively in a total of 54 patients from two published studies by the same research group [44,45]. Thyroplasty type II (midline lateralization) procedure was performed for the first time by Isshiki et al. [46] in a single female patient with adductor LD with restored voice with no recurrence of symptoms 17 months postoperatively. Nomoto et al. [40] compared the TA muscle myectomy and thyroplasty type II in 65 adductor LD patients, concluding that the TA muscle myectomy tends to improve strangulation, interruption, and tremor on the VHI scale [47]; however, postoperative VHI scores did not differ between the TA muscle myectomy and thyroplasty type II procedures [40].

## 6. Outcome Measures in the Treatment of LD

Acoustic and auditory-perceptual measures are used in evaluating the outcome of botulinum toxin injection in laryngeal muscles and speech and language rehabilitation as standard clinical procedures, as well as in evaluating surgical treatments of LD. Phonatory tasks include words or sentences having voiced phonemes/consonants or unvoiced phonemes/consonants (depending on the LD type) and the production of sustained vowel/a (termed often as “phonation”). Acoustic measures frequently analyzed include the following measures: jitter, shimmer, harmonics-to-noise ratio (HNR), smoothed cepstral peak prominence (CPPS), and a number of phonatory breaks. Standardized instruments include the use of VRQOL [31,48], OMNI-VES [32], USDRS [33], VHI [47], and Consensus Auditory-Perceptual Evaluation of Voice [49].

The assessment of voice quality can be evaluated with clinically available acoustic tools, such as newly developed VOXplot software (Version 2.0.1) (Lingphon, Straubenhardt, Germany, https://voxplot.lingphon.com/en/download/) (accessed on 27 October 2023.) [50,51,52,53]. Furthermore, recently, Hlavnička et al. [54] developed a robust automated method to allow precise tracking of multiple vocal tremor frequencies in 240 patients with different neurological diseases. It might be interesting in the near future to apply the developed algorithm by Hlavnička et al. [54] in LD patients since vocal tremors can also be present in LD patients. Future investigations could determine the psychometric properties of voice measures in LD in different language settings. Furthermore, better advancements in voice acoustic analysis, with the development of sensitive voice and speech samples of perturbed voice in LD, are highly needed for clinically distinguishing the most frequent LD types.

## 7. Genetic Studies in LD: The Missing Puzzles and Future Perspective

LD presents a paradigmatic challenge for genetic researchers. While clinical manifestations of LD have been well-documented, the genetic underpinnings remain enigmatic, with only a subset of cases attributable to known genetic causes. This gap in understanding serves as a puzzle piece in our comprehension of LD’s genetic landscape.

Historically, genetic studies in LD have been largely confined to candidate gene approaches, targeted gene panels, and microarray-based studies. While these have provided valuable insights into potential risk factors, their scope remains limited. Notably, about 25% of LD patients have a familial dystonia background, while around 12% of patients have a family history of other movement disorders. However, standard linkage studies in LD face challenges due to the scarcity of large family samples, inconsistencies in phenotype among affected relatives, dystonia’s low penetrance, and its late onset. Consequently, the specific gene mutations causing LD remain elusive [1]. Genes commonly tied to broader dystonia syndromes, such as DYT1-TOR1A, DYT4-TUBB4A, DYT6-THAP1, DYT25-GNAL, and DYT28-KMT2B, have also been linked to LD. Mutations in *THAP1* exhibit a broad clinical spectrum. Common presentations include oromandibular, craniocervical, and LD, but the literature also describes focal limb dystonia and segmental and generalized dystonia [55]. The *TUBB4A* gene missense mutation is correlated with a specific “whispering dysphonia”. Its clinical manifestations can vary widely, from isolated LD to severe generalized dystonia, accompanied by a distinct “hobby horse” gait [56]. Some mutations in the *TOR1A* gene cause early-onset generalized dystonia and were found to be associated with LD, which provided insights into possible common pathways in different types of dystonia. Remarkably, one single case of focal adductor LD with the *GNAL* mutation—without other dystonia types, a dystonia family history, or other movement disorders—has been recorded. This highlights the potential genetic overlap between LD and other dystonia types and suggests that mutations might account for isolated LD instances due to reduced penetrance. It is speculated that distinguishing truly sporadic from familial cases might be indistinct until focal LD-specific causative gene mutations are uncovered [57]. The missing pieces in LD genetics can potentially be found in rare variants, which might be responsible for a significant portion of the genetic risk and are often bypassed by standard genotyping methodologies. Traditional genotyping methods are not adept at capturing these rare variants, especially if they lie in noncoding or regulatory regions of the genome. Hence, whole-genome sequencing (WGS) emerges as a critical tool. With the capability to provide a comprehensive view of the entire genomic landscape, WGS can identify not only rare single-nucleotide variants but also larger structural changes and variants in non-coding regions. Kumar et al. [58] conducted the first study to use WGS to diagnose diverse dystonia phenotypes. Out of 111 participants, 12% were provided with a genetic diagnosis. The likelihood of a genetic diagnosis was higher in patients with earlier disease onset and combined dystonia phenotypes.

Additional advances in genetic research on LD aim to discern the polygenic risk that influences the manifestation of the disorder. Even though traditional genome-wide studies may be underpowered due to limited DNA samples from LD patients, one study even performed a polygenic risk score of LD and found a significant number of genetic variants lying near genes associated with synaptic transmission and neural development [59].

The future of genetic studies in LD seems poised for transformative discoveries, especially with the integration of advanced sequencing technologies and bioinformatics tools. The next frontier in LD research will likely involve multi-omics approaches, combining genomic data with transcriptomic, proteomic, and metabolomic insights. This will provide a comprehensive understanding of how genetic changes manifest in cellular dysfunctions and, ultimately, clinical symptoms. Moreover, the increasing emphasis on patient-centric research and collaborative international initiatives will aid in amassing larger cohorts for studies. Given LD’s rarity, multi-center collaborations will be crucial for achieving the sample sizes needed for robust genetic discoveries.

## 8. Discussion and Future Directions

In summary, two research groups reported neurophysiological findings on cSP recorded in TA [18,19] and CT muscles [21,23] in LD and healthy control subjects. The results indicated a shortened duration of the cSP in TA muscles in adductor LD [19] and altered cSP duration in a single case study with abductor LD [22]. From this perspective, an interesting point could be a future investigation of cSP in TA and CT muscles in each patient to verify intracortical inhibitory processes in individual laryngeal muscles due to different innervation of these muscles; CT muscle innervated by the superior laryngeal nerve and TA muscle innervated by the recurrent nerve. Currently, from the available literature, a single study reported norms for cSP recorded in CT muscles in healthy subjects [21], and no norms were published for cSP recorded in CT muscles in LD patients. Also, cSP recorded in the TA muscle was not reported in a bigger sample of individuals with abductor LD, except for a single case study reported by Rogić Vidaković et al. [22]. Investigation of norms of cSP duration in affected (target muscles) and non-affected muscles is an important step to understanding the neurophysiologic mechanisms of LD and possible compensatory strategy in the muscles non-affected by dystonia [26,28].

Regarding the new neuromodulation rTMS and laryngeal vibration treatment findings in LD, single rTMS sessions were feasible in mapping the M1 for laryngeal muscle representation in LD, with some beneficial effects on voice symptoms [11], while feasibility study using laryngeal vibration indicated that vibrotactile stimulation modulates neuronal synchronization over sensorimotor cortex and induces short-term improvements in voice quality in 69% of LD individuals [12]. However, randomized studies with sham-controlled trials could disentangle potential clinical usefulness or rTMS in LD. Also, the long-term effects of vibrotactile vibration are also unknown and require further systematic research.

Furthermore, DBS of the ventral intermediate thalamic nucleus pointed to beneficial effects on voice in subjects with essential tremor and LD. Currently, no studies have reported DBS effects in focal LD patients. Furthermore, ENT surgical procedures, thyroplasty type II, and TA muscle myectomy provide similar findings in reducing voice symptoms in LD [40]. Potential limiting factors of surgical procedures are smaller patient sample sizes and few surgical centers treating LD. The question that arises when choosing a potential surgical option is whether LD is a neurosurgical or an ENT problem. This question arises due to the fact that we still do not have a clear understanding of the patho-neurophysiological mechanisms that lead to LD disease.

Lastly, while the genetic studies in LD have left some puzzles unsolved, the rapid advancements in genomics and multi-omics approaches, combined with increasing global collaboration, offer a promising future. These efforts will not only elucidate LD’s genetic architecture but also pave the way for precision medicine interventions tailored to individual genetic profiles.

### New Approaches for Future Research

Future research could apply synergistic approaches by combining different methods to further understand the neurophysiological basis of different types of LD in relation to genetic research. For instance, laryngeal vibration therapy may have potential benefits in promoting laryngeal muscle function and coordination. Combining it with rTMS, which modulates neural activity in the motor cortex, could potentially have synergistic effects. Laryngeal vibration might enhance the effects of TMS by providing sensory input and facilitating motor learning. The effectiveness of combining these therapies would likely depend on the specific characteristics of the individual with LD and the underlying mechanisms contributing to their symptoms. A comprehensive treatment plan may involve multiple interventions, speech and language therapies, medical treatments (e.g., botulinum toxin injections), and possibly neuromodulation techniques. The specific combination of different methods for treating LD would need to be investigated through rigorous research approaches.

Finally, future TMS studies would need to further investigate the relevance of subclinical neurophysiologic markers such as cSP in the diagnostics and treatment of LD subjects.

## Figures and Tables

**Figure 1 brainsci-13-01591-f001:**
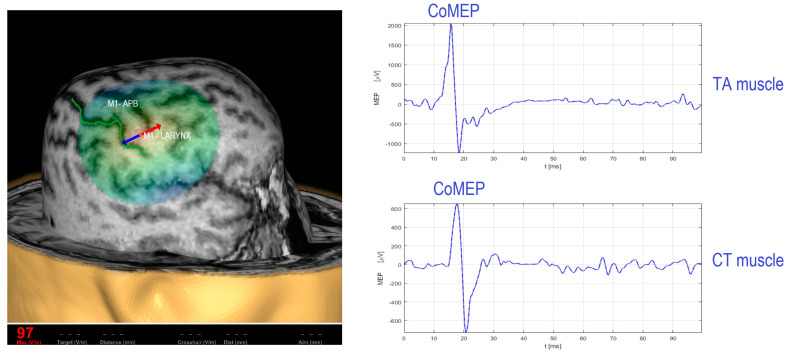
TMS over the M1 for laryngeal muscle representation (red-blue arrow) and CoMEPs recorded from TA and CT muscles. Left side: The central sulcus is depicted with a green line. Positive M1 referent spot for upper extremity representation (M1-APB). Legends: APB, abductor pollicis brevis; CoMEP, corticobulbar motor evoked potential; TA, thyroarytenoid; CT, cricothyroid. The magnetic stimulation onset is represented as the zero time. The time on the x-axis is expressed in milliseconds (ms), and the MEP amplitude on the y-axis is expressed in microvolts (µV). Source: single subject; ownership of the authors of the study.

**Figure 2 brainsci-13-01591-f002:**
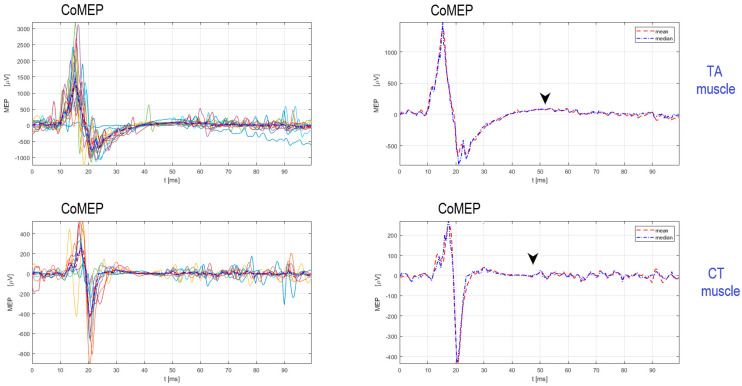
TMS over the single M1 cortical spot for laryngeal muscle representation elicits reproducible CoMEPs followed by cSP in TA and CT muscles. Note: Fifteen trials (different colours) from TA and nine trials (different collors) from CT muscles are superimposed with mean (dashed red line) and median (dashed blue line). The black arrow pointing down represents the offset of the cSP response, while the magnetic stimulation onset is represented as the zero time. The time on the x-axis is expressed in milliseconds (ms), and the MEP amplitude on the y-axis is expressed in microvolts (µV). Source: single subject; ownership of the authors of the study.

## Data Availability

Data are available with a granted proposal upon reasonable request.

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
