# Peer review of "Exploring Neurophysiological Mechanisms and Treatment Efficacies in Laryngeal Dystonia: A Transcranial Magnetic Stimulation Approach"

_brainsci, 2023, doi:10.3390/brainsci13111591_

Round 1
Reviewer 1 Report
Comments and Suggestions for Authors
I am pleased to review such an interesting paper; the review is well organized, and the data reported are very interesting. However some points need to be assessed or introduced before the publication:
- Botulinum Toxin is the main therapy for LD. There are no references regarding this therapy in line 71. I recommend the authors improve the introduction by implementing a brief paragraph on the role of botulinum toxin and adding some references.
- The authors reported essential tremor and tension muscle dysphonia. Please add some citations.
- I recommend the authors introduce some figures depicting the technique (stimulation and signal registration) to evoke COMEPS (and add some tables reporting normal values for cSP and latencies/amplitude for COMEPS). This is also true for the DBS and ENT procedures.
- Please explain the meaning behind the prolongation of the cSP and the increment in the latencies of COMEPS.
Author Response
The author's response is attached to a PDF file.

Reviewer 2 Report
Comments and Suggestions for Authors
This manuscript presents a scoping review of recent findings in laryngeal dystonia.
There is some confusing in the organization of the paper given that is the purpose – for example, the first section is discussing the author’s previous work which is in healthy people. A more logical organization would be to start the section discussing findings related to impaired intracortical inhibition, as that related to the purpose of the paper, and perhaps later discuss some mechanics of TMS. However, the mechanics of TMS is not really discussed either, for example, If the authors wish to discuss TMS in detail, they should highlight different approaches for electrode placement and the cSPs should have an indication of onset/ offset. The figure is rather vague in terms of how the cSP is quantified.
Relatedly, the phrasing in section 2 is awkward in that the topic sentence appears to state that the authors developed all TMS laryngeal research methodology. It could be interpreted as self-promoting, particularly because it doesn’t related to LD.
Throughout the paper also lacks and statement of the quality of research – it seems to qualify all results similarly without regard for quality of the work (ie, single case vs larger study, etc). Some level of interpretation and critique would be appropriate. For example, the reviews of the surgical interventions should mention limitations of the studies to give some context to the reader.
Similarly – in the assessment section, what is the consensus of optimal measures to report?
The references are not always logical or up to date. For example, in the neuromodulation section they mention a small study but then cite a review that is nearly 20 years old that is not specific to LD (ref #18).
In the future directions section – give context to why these would be good additions. Why are norms for cSP needed?
Minor:
Line 93: loaded with a pair of fine-wire hooked electrodes (#019-772800, 93 Nicolet Co., Middleton, WI) was used. Phrasing – don’t need ‘was used’
Comments on the Quality of English LanguageNA - primary issue is organization and detail
Author Response

(The authors gave the same response as above.)
